

# Identifying urban areas prone to flash floods using GIS - preliminary results

Marzena Wicht[1] and Katarzyna Osińska - Skotak[1]

[1]Warsaw University of Technology, Faculty of Geodesy and Cartography, Department of Photogrammetry, Remote Sensing and GIS, Pl. Politechniki 1, 00-661 Warsaw

*Correspondence to:* Marzena Wicht (mwicht@gik.pw.edu.pl)

**Abstract.**

The aim of this study is to develop a consistent methodology to determine urban areas that are particularly vulnerable to the effects of heavy, rapid rains. They are, as a result of climate change, more and more prevalent in the temperate climate, usually spring - summer from mid-May to late August - and involve the risk of flash floods. In recent years, the increase in incidences

of such phenomena is noticeable throughout the whole of Europe. It is assumed that through the analysis of environmental and infrastructural conditions, using the developed methodology, it is possible to determine areas vulnerable to flooding due to torrential rains. This may lead to a better management, quicker response in case of a phenomenon, and even preventative measures to reduce the occurrence of adverse effects of torrential rains (for instance modernisation of the urban drainage system and development of methods to get rid of rapidly collected water). Identifying areas particularly vulnerable to the effects

of heavy rains can be achieved by adapting hydrological models, but they require an appropriate adjustment and highly accurate input data: (based in situ or radar measurements of precipitation, land cover, soil type, humidity, wind speed, vegetation species in a given area, growing season, the roughness and porosity of the cover and soil moisture) but such detailed data are generally hard to obtain or not available for less developed areas. It could also be achieved by performing spatial analysis in GIS environment, which is a more simplified form of modelling, but it gives results more quickly and the methodology can be

adapted to the commonly available data. The proposed methodology was tested in Warsaw's central sub-district - Powiśle.

**Key words:** GIS, urban areas, flash floods, urban floods

## 1   Introduction

As violent weather events occur more and more often in the North Temperate Zone across Europe (Gaume et al., 2009), certain

measures to handle their outcomes or to mitigate their effects have to be taken. In the recent years the number of flash floods has risen significantly (Huntington, 2006), and Poland is yet another country that suffers from this intensified phenomenon (Instytut Meterologii i Gospodarki Wodnej, 2016). In these floods, urban areas are particularly vulnerable to its effects, as the share of impervious surfaces is substantially higher than in rural areas. Streets, pavement and other impervious areas usually increase the speed of the run-off (Banasiak, 2012), which often results in more severe damages and presents a valid threat to

the inhabitants.



There are several different indicators, that influence the vulnerability of a given terrain to the flash floods, such as: climate,topography, geomorphology, drainage capabilities and man-made structures (Youssef et al., 2011). There have been many different approaches throughout the years to capture and describe the drainage possibilities, and through it, the susceptibility to exposure to the effects of heavy, rapid rains. These include conventional geomorphologic approaches (as presented by Horton
(1945), Leopold and Miller (1956), Strahler (1964) or in Krishnamurthy et al. (1996)), where researchers focus on the basin's geometric and relief characteristics. However, they all argue that it is cost- and time- extensive to examine all the drainage networks (especially in a large area, from in situ measurements) due to their complexity. Therefore, for some years now, an automatic DEM extraction from many different sources (satellite imagery, airborne laser scanning, photogrammetric point clouds) aids drainage networks and basin extent extraction (Ozdemir and Bird, 2009). On the other hand, many hydrological
models have been developed and are nowadays successfully used to represent possible outcomes of the flash floods. They require high resolution input data and derive highly detailed outcomes. Unfortunately, most of these models are incorporated to the expensive software and are not user-friendly for the decision makers (local governments, non-hydrological experts). The more accessible and rapidly increasing, open-access platform has recently been GIS. In most of the western European countries, GIS is widely used in municipalities' departments that deal with surveying, cadastre, spatial planning or environmental
protection (Arnaud (1993), Mączewski and Staniewska (2005)). Using the GIS tools to derive geomorphological information required for flash flood has been already a subject of other studies (like Youssef et al. (2011), Klok (2012) or Abdelkareem (2016)). However they have not performed vulnerability assessments and have not taken some of the crucial indicators into consideration. Researching urban areas requires particular attention to the existing drainage sewage system, soil features (structure and physical information) and building information. This research aims to develop a consistent methodology of mapping urban
areas, that might be prone to the effects of flash floods and takes into consideration above mentioned factors. These information should be available in any European municipality, therefore such analysis can be performed by any local government which might be endangered by flash floods. Short overview of the selected European sources of input data essential for discussed methodology provides Table 1. Gathered information was based upon available data provided by country's geoportals and INSPIRE Directive (Directive, 2007). It is clear, that more detailed data (such as detailed land use, soil map or sewage system)
have to be obtained at the local level, nevertheless Table1 provides insight in possible data sources and confirms thesis, that proposed methodology can be implemented in European cities.

The steps of the research are as follows: a) develop a simplified GIS methodology to map urban areas prone to flash floods; b) test the thesis in different European urban areas; c) perform input data sensitivity analysis; d) develop user-friendly software to perform flash flood mapping in different municipalities. The research presented in this paper presents the design of the
methodology, as well as tests its functionality on a case study in Warsaw.

Over the last decade, Warsaw has suffered a couple of flash floods caused by heavy summer rains (particularly harmful in 2010,2011 and 2016). One of its sub-districts, Powiśle (see Figure 2), sustained particular damage due to it's terrain model - Warsaw's escarpment runs through its boundary and provides up to 12 m height difference. On the other side Powiśle is limited by the Vistula River. A detailed overview of the terrain model can be seen in Figure 6. Furthermore, its location in the heart of



**Table 1.** Possible data inputs in selected European countries.

Data available free of charge / *Data available on demand for scientific purposes* / **Fee applies.**

| Country | Land Use | DTM | soil | rain |
|---|---|---|---|---|
| Austria | CLC2012 | DTM 10 m | FAO Bodentypen | **radar/rain gauged** |
| Belgium | CLC2012 | *LIDAR, 1 m* | 1:20 000 | radar/rain gauged |
| Bulgaria | LCC on Landsat 30 m, cities 0.7m or 1m | **LIDAR** | on demand at Research Institute of Soil Science and Agroecology | *NIMH / rain gauged* |
| Croatia | LCC on ortophoto | *LIDAR* | CGS: Basic Geological Map of the Republic of Croatia 1:100 000 | *Croatian Meteorological and Hydrological Service / radar/rain gauged* |
| Czech Republic | CLC2012 | DTM 5m | 1:250 000 | Czech Hydrometeorological Institute/radar |
| Finland | topo 1:500 | LIDAR | 1:250 000 | The Finnish Meteorological Institute / radar |
| Germany | *ATKIS* | *LIDAR* | *1:50 000* | *DWD / radar* |
| Netherlands | *Land Use database* | *LIDAR* | **Bodem arten** | radar/rain gauged |
| Poland | *base map (1: 500) paper/digital* | *LIDAR* | *1:50 000* | IMGW / rain gauged / **radar** |
| United Kingdom | *LCC 2012* | *10 m Contour* | *1:250000* | *radar/rain gauged* |

Warsaw, a neighbourhood, where new, luxurious housing investments peak out, makes it even more prone to the damages and is worth investigating.

## 1.1 Flash floods in urban areas

Annual extreme weather events cause fatalities and contribute to economical damage (Bednarczyk et al., 2006). The economic
5  loss due to flash floods are one of the most considerable natural hazards and concern hydrological, climate or natural hazard sciences due to their frequency, amount of people affected on a global scale and actual fatalities (Marchi et al., 2010). As reported by Barredo (2007), between 1950 and 2005 almost half of all casualties related to floods are appointed to flash floods. To emphasize the message, economical damages have to be taken into account. A single flash flood event in several southern French regions caused an estimated €1.2 billion economic damage in 2002 (Huet et al., 2003). Danger to both property and
10  life makes flash floods even more concerning, although with increased pressure on land use, it makes the issue even more difficult to resolve. Moreover, as (Huntington, 2006) argues, the global hydrological cycle is intensifying, which can be proven by evidences of increased precipitation in both, global (Groisman et al., 2005) and continental scales (Klein Tank and Können, 2003). Due to repercussions of the global change on climate and natural phenomena (U.S. Geological Survey, 2014), flash





flood hazards are expected to occur more often in the future (Marchi et al., 2010). As urban areas mostly consist of impervious surfaces, they are even more prone to the negative effects of flash floods. These floods are usually associated with short and intensive precipitation and mostly impact small basins (less than 1000 km$^2$) (Marchi et al., 2010). Time occurrence is connected to a) size of the catchment and b) surface run-off. Run-off is influenced by several different factors,such as the magnitude of

rainfall, sewage system or soil moisture. Furthermore, it is often intensified by land use modification and urbanisation (Loaiciga et al., 1996). Diligent characterization of affected areas may provide better insight to understand the processes occurring during these extremes, which may lead to better risk management in the future. Even more so, since warning and emergency management in flash floods forecasting is particularly challenging. Due to short time response and often insufficient data (both temporal and spatial resolution) the forecast are usually too late or inaccurate (Norbiato et al., 2008). However, many attempts

have been made (Drobot and Parker (2007), Norbiato et al. (2008), Ntegeka et al. (2015)) to improve forecasting (even by using social-media response as input data (Koole et al., 2015)). Nevertheless, modelling such phenomena, regardless of the discharge magnitude, can only improve profits of preventing the occurrence of negative effects caused by flash floods (Radoń and Piórecki, 2012).

In the recent years, research in urban flood modelling intensified. Not only the researchers, but also local government
have seen the negative outcomes of climate change, average temperature rise and accelerated hydrological cycle. These put a significant emphasis on the importance of detecting and modelling upcoming extreme weather events. According to Welle and Birkmann (2015), flash floods,among others, account for 81% the most frequent and devastating natural hazard and 83% of connected fatalities. Whereas it is known, that the biggest devastation occurs in less developed countries, the significant damage is also taken by western countries, especially cities and urban areas. This resulted in numerous studies investigating
the possible outcome and modelling of flash floods, not only in hydrological sciences, but also using GIS techniques. De Roo et al. (1996) proposed openLISEM, which models the surface water and sediment balance for every raster cell. It is a open-source model, however it produced the output maps ins PCRaster format, not supported by OGC. What is more it does not consider the sewage system and has not been widely used for urban applications. WOLK, as described in Klok (2012), is also grid based overland flow computation model and takes DEM and precipitation as input data. Here however, the model resigns
from many other possible input parameters, like soil type, (im)pervious surfaces, sewage system or different routing algorithm. Also, acording to Klok (2012), WOLK cannot handle bigger datasets, it's time-expensive and is limited to one directional routing. The approach presented by Abdelkareem (2016) used GIS techniques and remote sensing data to discover areas particularly vulnerable to flash floods, but the research focused more on several morphometric parameters of basins, like basin relief, texture, and geometry and until now was only tested in the rural area. Chen et al. (2009) have developed similar model
to simulate urban flood inundation (GUFIM), and they use as an input data contour maps and non-detailed underground sewer system data. However, they also analyse the initial status of soil moisture (which in many cases is simply not available) and sewer system conveyance rate, as opposed to the actual localisation of inlets and pipes. Nevertheless, a thorough benchmarking of these discussed models is planned in order to compare the results with the proposed methodology. The advantage of our approach is directed strictly to the urbanised areas, which in recently accelerating urbanisation only finds its application. It
takes into account various factors, which are present in the urban environment: sewage system, which at the beginning takes




upon some drainage, but crossing certain threshold might be the cause of re-flooding; open soil and vegetation areas, which slow down the surface runoff and are able to take over some of the precipitation; complicated and mostly connected impervious surface structure, which is much more present in urban areas than anywhere else. It is also worth mentioning, that model uses the data, which are commonly available at the average municipality level in Europe and is open-source based, thus might be widely used by the local governments. This particular study investigates possible outcomes of a flash flood in Warsaw and uses the data available at hand from the council office, however in our future works, we also want to explore available datasets in other European municipalities.

## 1.2 Study area description

While modelling itself is the core of dispute, a key to properly adjusting methods is to know and understand the modelled terrain. The study area is situated within the central Warsaw district - Powiśle. Its location between the Warsaw escarpment and Vistula river, making it even more interesting to investigate. The study area occupies approximately 327 ha. As mentioned above, Warsaw's escarpment is a fundamental topographic element of Warsaw, shaping the landscape and spatial structure of the city. Within the study area, the escarpment achieves its biggest height differences (varying from 8 to 22 m) (Pluta, 2008), which, in case of a flash flood substantially accelerate the run-off. The escarpment was one of the key factors for the location of Warsaw. Positioning castles and later residences at the slope had many advantages: easier defence, attractive views, favourable climate (ventilation through the valley of the Vistula), dry ground and the lack of flood danger - as opposed to areas located in the valley - which are mostly the study area.

Besides the topological features of the terrain, when potential precipitation is considered, geological features also play crucial role. To fully comprehend the infiltration and run-off capability of the investigated area, those characteristics must be recognised. In this particular case, Powiśle is comprised primarily of anthropogenic formations. The geological structure of the terrain is the result of Riss glaciation (Habbe et al., 2007). The shallowest glacial sediments layer of the terrain is covered with tracks of seven hundred years of human activity - thus it is hard to recognize the formations which are older than anthropogenic. These have been created over hundreds of years due to littering, but not exclusively. Preceding the Second World War, this area used to be a industrial district - for ages Vistula river served as a trade and transport route, which resulted in a number of factories and poor, unregulated labourer cot houses (Paździor and Wierzbicka, 1994). The Second World War brought destruction to almost the whole city, together with Powiśle district. What once used to be a bustling place, was transformed into post-modern architecture feast filled with vast parks (see Figure 1). The crucial information here, however, is a fact, that most of the rubble left after the bombing throughout the course of the Second World War was used to rebuild and level the area. This significantly influenced the infiltration capability of the area - higher soil porosity results in faster and bigger infiltration volume (Kuźnicki et al., 1979) as well as strongly influences landslide possibility (Święcicki, 1981), which also (Paździor and Wierzbicka, 1994) turned out to be the issue in the study area in the past.

Some part of the analysed area is covered by vegetation located on the escarpment, in the form of parks or as lawns -, approx. 40% of the terrain is occupied by pervious surfaces. Run-off will therefore be partially decreased by vegetation. In case of flash floods, run-off occurs when the "infiltration of water in the soil profile is less than the amount of water that received the soil



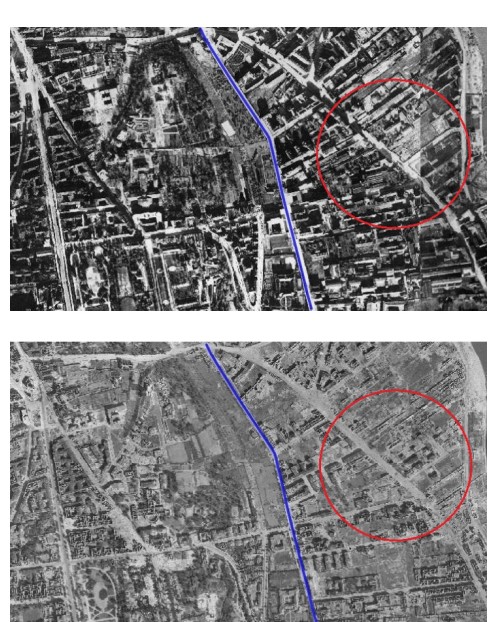

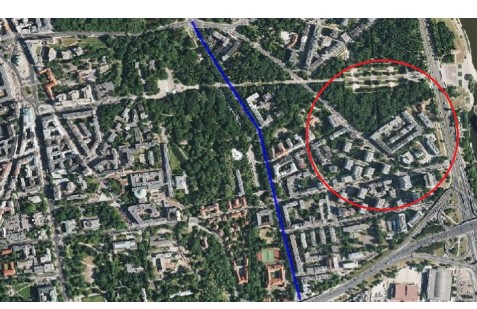

**Figure 1.** Aerial photographies of part of study area respectively in 1935, 1945 and 2008. *source: http://www.mapa.um.warszawa.pl/*

surface" (Święcicki, 1981). It depends on the slope surface, vegetation, soil structure and on their permeability, along with other factors (Norbiato et al., 2008). Since most soils in the analysed area are the anthropogenic soils, it is difficult to define clearly what amount of water would be absorbed by the soil. The study area is covered by soils transformed mechanically, including as a result of the war, i.e. bulk soils, silicate - debris and rubble - as mentioned above. However, the rubble is likely

5  made up of concrete or brick. In this case, such brick returns over some time to its original form - clay (Kuźnicki et al., 1979). It is therefore difficult to clearly determine the porosity of the soil in this area, and thereby the water capacity as well.

Average porosity soils of different origin are as follows: a) sandy soils - 35-45%; b) loam and loess - 40-50%; c) clays and silts - 50-60%; d) organic soil (peat) - 80-90% (Kuźnicki et al., 1979). Figure 2 shows that the largest part of the area is occupied by anthropogenic soils which mainly consist of embankments and sand. We can assume that the majority of these

10  anthropogenic soils are soil developed from river sediments. The porosity of such soil is in the range of 35 - 46%. Years of littering and adding a rubble, to a small extent has increased the porosity of the soil (Kuźnicki et al., 1979). It is therefore





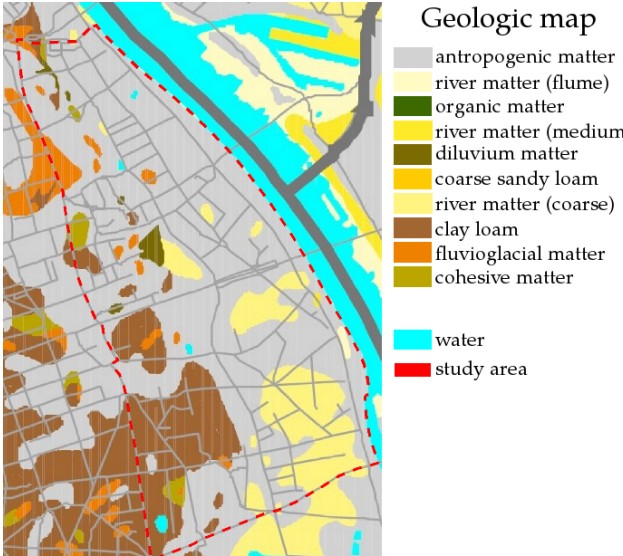

**Figure 2.** Geological map of the study area. source: http://www.architektura.um.warszawa.pl/ekofizjografia

necessary to average the porosity and the volume of the soil occupied by the solid phase. It can be assumed that approximately 50% of the soil volume is formed with soil pores of different diameters. Therefore, if 50% of the volume of soil consists of pores and 50% of the pores are filled with capillary/gravital water or water vapor, then the rest, less than a quarter of the soil volume, consists of medium and large pores, which are air-filled and provide a reservoir for water absorption. This means, if 100 ml precipitation occurs, approx. 25 ml is absorbed by the soil, while the rest will create run-off (Święcicki, 1981). Such brief analysis of the terrain enables to draw conclusions about geological features which will prove valuable at the later stage of analysis.

## 2    Data and methods

### 2.1    Precipitation

The Institute of Meteorology and Water Management (IMGW) acts as the metrological service in Poland. Its tasks include, among others, development and broadcast of forecasts and warnings for the population, including warning against storms and flooding as their result. IMGW leads the National Hydrological and Meteorological Service, which provides information about the state of the atmosphere and hydrosphere and therefore the possible outcomes of disturbances within them. IMGW administers precipitation measurement stations as well as radar weather stations across Poland. As radar data are not available for public use, only the precipitation measurements provided by rain gauges are a valid source of information. Although many stations are available for Warsaw (see Figure 3), IMGW shares data just from one station (which is approximately 8 km south-west from the study area). Furthermore, data are being published as daily summaries of precipitation. Taking those facts into





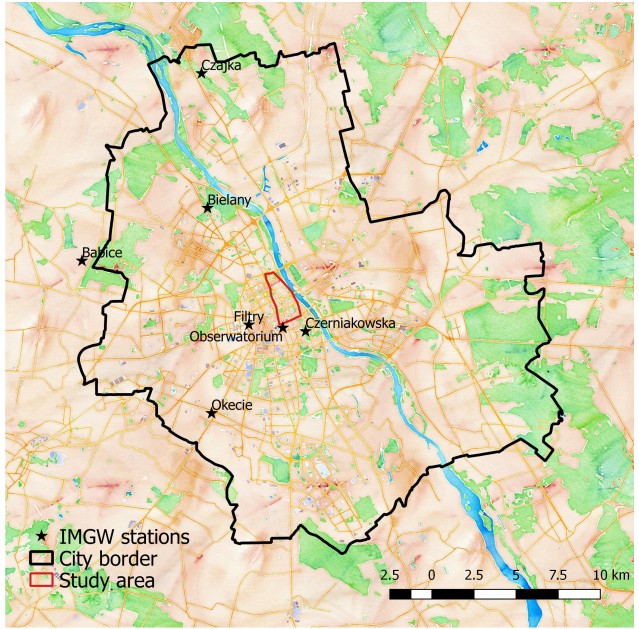

**Figure 3.** IMGW meteorological stations in Warsaw.

**Table 2.** Monthly maximums [mm/m$^2$] of daily precipitation in Warszawa - Okęcie measurement station.

|        | 2007 | 2008 | 2009 | 2010 | 2011 | 2012 | 2013 | 2014 | 2015 | 2016 |
|--------|------|------|------|------|------|------|------|------|------|------|
| May    | 9,0  | 10,0 | 16,6 | 27,0 | 14,0 | 17,8 | 46,4 | 25,4 | 16,7 | 16,4 |
| June   | 28,8 | 8,2  | 28,6 | 23,4 | 10,0 | 15,0 | 19,0 | 18,4 | 5,1  | 21,5 |
| July   | 11,6 | 27,0 | 28,0 | 29,8 | 75,8 | 18,0 | 11,0 | 29,4 | 15,3 | 35,4 |
| August | 21,2 | 36,6 | 16,4 | 39,2 | 26,2 | 7,4  | 27,4 | 19,8 | 6,4  | 21,7 |

consideration, it is safe to assume, that no spatial or temporal resolution term can be used here. Nevertheless, this study aims at the countries and areas which are less developed and/or do not have all the necessary data required to perform standard hydrological modelling. Therefore, the summer (May-August) epochs between 2007 and 2016 were analysed (see Table 2). For these epochs, a maximum of 39,2 mm/m$^2$ (2010) was chosen as input data of precipitation.

5  ## 2.2  Digital surface model data

The early steps of the modelling began already in 2011. Unfortunately, airborne laser scanning data was not available at that time for all areas in Poland and back then, also not free of charge for scientific purposes. Therefore, an analogue base map for the terrain was acquired (scale 1:500, 42 sheets), updated (only on buildings level) and digitized together with 14849 elevation points existent on the map. These data were then used to create the DTM of default 8 m (point density prevented from creating

10  a more detailed outcome), it was then re-sampled to the resolution of 5 m. Today, as LiDAR data are available free of charge



for scientific purposes and vast areas of Poland are covered (Centralny Ośrodek Dokumentacji Geodezyjnej i Kartograficznej), DTMs can also be obtained with much better resolution and have already found their application for hydrological modelling (Meesuk et al., 2015). During the preliminary trials, which this paper presents, we used DSMs obtained from surveyed elevation points, as described in Section 4. Currently, we have managed to obtain LAS data and we are carrying out tests based on them.

## 3 Design of experiment and methodology

### 3.1 Digital Surface Model (DSM)

The main assumption of the proposed methodology lies within the comprehension of the area. As geologic and topolgic features of the area have been briefly discussed, a proper way to represent it should be chosen. Flash flood modelling has become a topic relatively recently, since modelling in urban areas is far more complex compared to the rural areas. Another reason for that is, that until recently topographic data of adequate resolution and accuracy have been missing. Nevertheless, digital surface models (DSM) obtained from airborne scanning laser altimetry - LiDAR are becoming available for many countries and regions in Europe and some (with coarser resolution) can be downloaded free of charge. LiDAR data, with appropriate point density is able to deliver sub-meter spatial resolution (Baltsavias, 1999) and therefore allows to properly represent the complexities of urban topographies. Hydrological science has been using digital terrain models (DTMs) obtained from LiDAR for many years now (Mason et al., 2007) and quite often are parametrized using them as model bathymetry (Marks and Bates, 2000). However, there are drawbacks to it. This technique provides lots of points, with one to few points per square meter and height accuracy of less than 15 cm (Bakuła, 2011), thus is responsible for the high redundancy. According to Bakuła (2011) it is a common problem for algorithms used in flood modelling, since most of the software restricts maximum number of points used for the calculation. Fortunately, with use of numerical calculations it is possible to derive a suitable DTM for flood modelling by generalizing it, without loosing its accuracy Bakuła (2011). Another disadvantage to LiDAR-based DTMs is the frequency with which campaigns are carried. Some countries, like the Netherlands have already completed a second campaign - first carried out in 2003 - (AHN, 2015) (and the third to be completed in 2019 is supposed to be initially free of charge), whereas some regions or areas, like Poland, have not finished the first one yet. For the purpose of this study it is assumed, that the study area is not big enough to require reduction of DTM, thus LiDAR-based DTM is suitable for the purpose of this research. However, we propose photogrammetric point cloud obtained, i.e. form a drone flight as an alternative source, should the topicality of the LiDAR campaign be not satisfactory.

### 3.2 Flow direction

The direction of flow is determined by the direction of steepest slope, or maximum drop, from each cell. It is obtained directly from DSM. There are eight valid output directions relating to the eight adjacent cells into which flow could travel (see Figure 4). This approach is commonly referred to as an eight-direction (D8) flow model and follows an approach presented in Jenson and Domingue (1988).





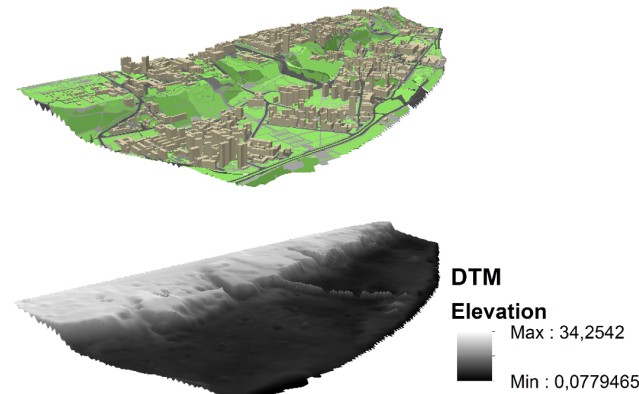

**Figure 4.** Flow direction coding. *based on ESRI Resource center)*

However, there are two approaches possible here - the single flow direction algorithm and multiple flow direction algorithm.

### 3.2.1 Single flow direction algorithm

This very common algorithm is used to determine topographical features of the terrain based on the assumption, that discharge can occur only into one cell at the time. However, there are few ways to approach that matter, as discussed in O'Callaghan and

5 Mark (1984) or Fairfield and Leymarie (1991) or in Lea (1992). Each cell is being compared to the elevation of its neighbouring cells and only one, the steepest slope is assigned to the investigated cell (Wolock and McCabe, 1995) What is worth mentioning, is a fact that flow direction is also written for 'flat' cells in order to provide flow path from all cells. Simply, the cell with the steepest slope is being found among the neighbours to the flat area, and the direction which it represents is assigned to the area (Jenson and Domingue, 1988).

### 10 3.2.2 Multiple flow direction algorithm

It uses two different types for calculation: one, for steep areas (cells with at least one downslope neighbouring cell) and second for flat areas. The base assumption is that all neighbouring cells may receive run-off, as long as they are situated below the investigated cell. It redirects the flow to multiple lower cells in proportion to the slope between them. However, different algorithms are used to calculate the proportion of the run-off: contour-based in (Wolock and McCabe, 1995) or in (Quinn et al.,

15 1991), non-contour based in (Freeman, 1991), (Holmgren, 1994), (Costa-Cabral and Burges, 1994) or in (Pilesjö et al., 1998) and other approaches like (Mitasova et al., 1996) or (Tarboton, 1997).

For the test of the methodology, single flow direction algorithm will be used, as proposed in Jenson and Domingue (1988), yet in further studies a detailed analysis of impact of different flow algorithms on the result outcome will be carried.





### 3.3 Flow accumulation

Flow Accumulation computes accumulated flow as complied weight of all cells flowing into each downslope cell in the output raster. "If no weight raster is provided, a weight of 1 is applied to each cell, and the value of cells in the output raster is the number of cells that flow into each cell" (Environmental Systems Research Institute, 2016). Our assumption was to apply a

weighted raster which takes into account (im)pervious surfaces. In order to detect these land covers, some simplification has to be used. The information whether the given area is impervious can be obtained from satellite/aerial imagery (middle resolution imagery is sufficient (Harvey and Hill, 2001).) with near-infrared band, by filtering out vegetation using the NDVI (Normalized Difference Vegetation Index). It functions on an assumption, that image is taken at a specific time of the year, when open soil (in urban areas mostly lawns or sport fields) is covered by vegetation. Naturally, should any green roofs be detected, those false

positives can be filtered out by utilising a buildings mask. Another possibility is to use already existent geodatabases with the information of specific land cover at a sufficient accuracy. The drawback to this method concerns the topicality of the data as well as availability of digital data in less developed areas. These approaches can be used simultaneously and aid one another to obtain the best possible outcome without necessity of investing in VHR imagery, which usually generates extra costs.

If assumed rainfall intensity is applied to each raster cell (i.e. 20 mm/m$^2$, raster cell = 2x2 m then each raster cell receives 80

mm), a certain amount can be discharged due to the infiltration capabilities, which were discussed in Section 1.2. Therefore weighting raster with appropriately lower discharge values for the pervious areas is applied in this step.

Another discharge possibility lies within the sewage system. If all the outlets are determined and flow capability od the drainage system is assessed, single outlet points in a form of raster cell can be designated as a certain discharge amount.

### 3.4 Basins

The drainage basins or catchment areas are found by identifying ridge lines between them. As local basins and sinks (the lowest cell in each basin) are found based on the flow accumulation and direction results, the total amount of rainfall for each basin is calculated. Creating such small local catchments enables better comprehension of endangered areas and finding the lowest cells (sinks) within them makes the next step possible.

### 3.5 Detection of endangered areas

Based on the flow accumulation output with applied weight, each raster cell represents accumulated water. Since within local basins all discharge accumulates in sinks, the precise amount of water (in litres) is known for each sink. Based on the water volume amount and DSM we can estimate how high the water rises from chosen cell (in this case the local sink).

Once the relative water height is estimated, a contour with its value is dealienated creating a geometry representing vulnerable areas within each basin.

With a fully automatised process we obtain a map of endangered areas for each local basin and with proper data input, also a information what and where exactly flooding occurs - a housing estate, a school or a hospital. This provides decision





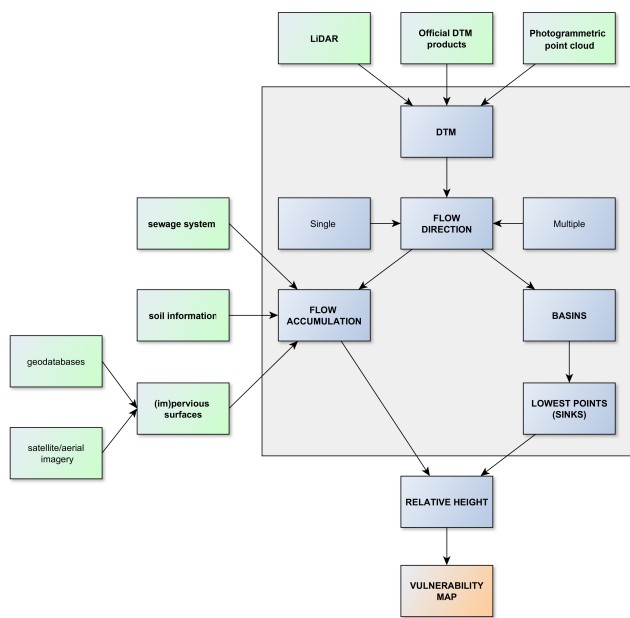

**Figure 5.** Model schematics: green - input data, red - output.

makers with a vulnerability assessments of a given area, and can be used during disaster management, i.e. while dispatching fire-fighters squads or informing the inhabitants.

Designed methodology is presented in the Figure 5.

# 4   Model implementation

As mentioned in the previous section, the first step was to obtain a DSM for the study area. There are no ready DSM products to obtain, therefore it was created based on available data. The first attempts to implement the proposed methodology have began already in 2011, when no LiDAR data were available, as mentioned in Section 2.2. The only feasible resources free of charge for the scientific purposes at that time were old, analogue city base maps in 1:500 scale. Altogether 42 base map sheets were georeferenced and digitalized (at the same time updated based on local vision and inventory, LOD level 1-2), together with 14849 surveyed elevation points. They were used to create DSM with a resolution of 8 m, and 5 m after re-sampling. (see Figure 6).

Based on that flow direction and flow accumulation was calculated (see Figure 7). The weighted raster for impervious surfaces has been applied during the flow accumulation step. Sewage system data was acquired for the study area, however due to the fact that all infrastructure lines are projected and maintained in computer-aided design and drafting software (CAD), the challenge lies in importing it to the geodatabase with the automatised attribute assignment (the lines in CAD software are captioned, not attributed). As Verlaat et al. (2015) suggests, it can also be obtained by removing some part of discharge amount

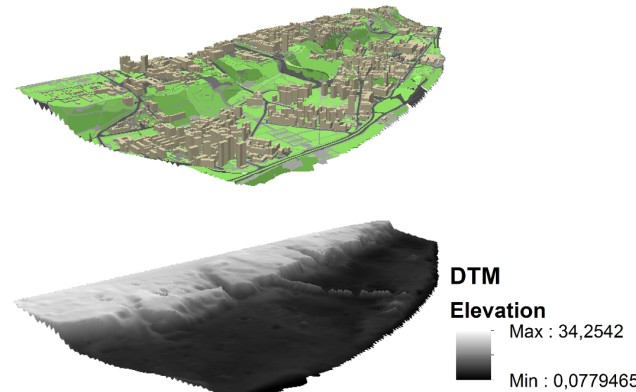

**Figure 6.** Study area in 3d view.

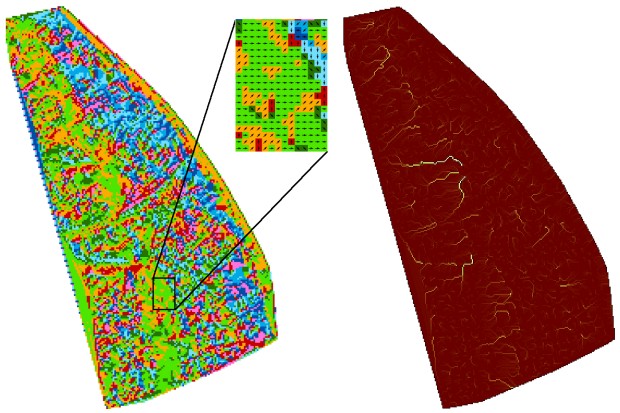

**Figure 7.** Flow direction and accumulation computation. Direction coding as in Figure 6.

(they propose 20 mm), which was done at this step of the research. It is planned in the future to add a georeferenced sewage system with the storm drains (inlets) localisation over the given diameter of a pipeline, to simulate and add supplementary drainage points to the flow accumulation weight raster.

   The next step was to designate local basins and lowest points in each basins. It included converting raster basins into vector and determining cells with accumulated flow in each catchment area. Unfortunately, due to the poor resolution of the raster, the creation of catchment areas resulted in artefacts, meaning catchments of the size of 1 pixel were created. They were therefore removed from further analysis. Figure 8 shows above described steps.

   After dealienating local basins and finding accumulated cells in each one, the total amount of water accumulated in each basin is derived. Values are classified into 5 classes (natural breaks algorithm) and the last three classes are appointed as potentially risky. The last step was to derive, based on water volume and DSM features how high will the water rise. Dependant on the




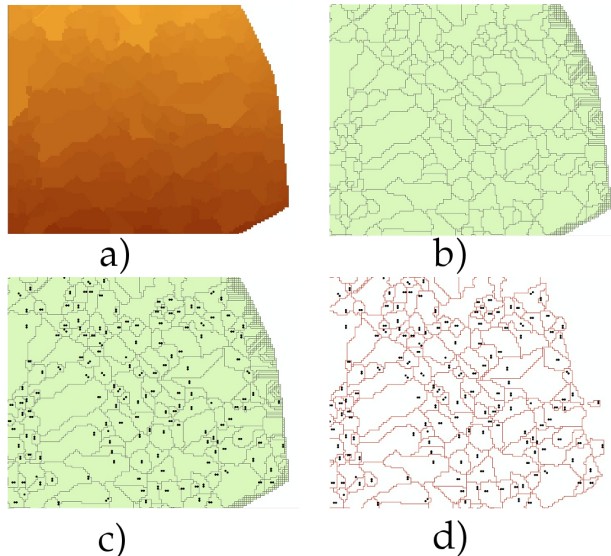

**Figure 8.** Local catchments computation steps; a)local catchments dealienation b)conversion into vector layer c)sinks identification d)artefacts removal

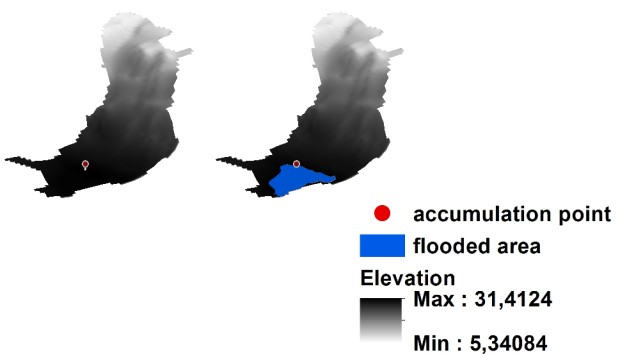

**Figure 9.** Deriving water height plane.

volume and surface input a reference plane is derived in a form of a contour which represents then flooded area, as seen in Figure 9.

# 5 Results

The implemented methodology derived 1864 local catchments with lowest accumulated cell in the study area. The volume
5   ranged from 0,14 m$^3$ to 952 m$^3$ of water. For the detailed analysis exemplary basins (see Table 3) representing three most





**Table 3.** Comparison of exemplary endangered basins.

| Basin no. | basin area [ha] | rainfall [m$^3$] | absolute height [m] | absolute height [mm] |
|---|---|---|---|---|
| 1 | 0,705 | 476,990 | 6,225 | 22,599 |
| 2 | 2,811 | 132,035 | 6,179 | 117,999 |
| 3 | 1,661 | 82,313 | 11,053 | 5,399 |
| 4 | 1,144 | 695,345 | 6,137 | 13,700 |
| 5 | 0,905 | 853,957 | 6,626 | 62,699 |
| 6 | 1,617 | 715,236 | 7,462 | 46,299 |
| 7 | 1,192 | 66,297 | 21,548 | 254,899 |
| 8 | 0,142 | 84,857 | 8,578 | 57,800 |
| 9 | 0,298 | 121,127 | 8,116 | 11,699 |
| 10 | 0,964 | 465,037 | 8,102 | 10,200 |
| 11 | 1,327 | 68,167 | 18,357 | 35,700 |

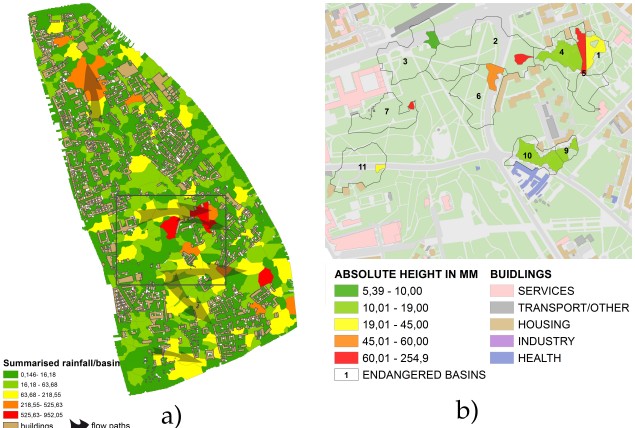

**Figure 10.** Basin classification (a) and derived over flooded areas (b).

endangered groups were investigated. Naturally, the least vulnerable areas are located at the top of the escarpment as well as parks, since no buildings or other valuable constructions are located there. Figure 10a presents the basic classification and Figure 10b detailed overview of the exemplary endangered areas.

Almost all of presented in 10 basins are located under the escarpment, which in this particular area creates the height difference of about 18-20 m. Fortunately, much of this area is occupied by the vegetation and open soil - parks and recreation areas. From the 10 b we can see, that absolute height of the water rise vary ro 5 mm to 254 mm. Since buildings have been masked out from the DTM (as the water does not flow through the building), water planes which are represented by 5 classes in 10 b do not cover the buildings, therefore no immediate danger is visible on the vulnerability map. Nevertheless, a large water





plane is observed just outside the W. Orlowski Clinical Hospital directly at the basin 9 and 10. Even the slightest risk of having hospital flooded means immense financial losses and puts the patients as well as the stuff in danger. In this particular case,a medical incineration plant as well as mortuary is located in a sub-basement level. In an event of flooding this floor, the possible outcome might endanger a much bigger area, as the water might be transporting then pollutants and infectious matter beyond

the hospital walls. Although the plane height depicts maximal height as 19 mm, it might be actually more, as the model might be underestimating the water accumulation. In order to perform detailed analysis, hydrological model should be run to verify the extent of damages. However, as previously mentioned, this model is only supposed to determine potential engendered areas to hasten the risk management decisions in case of the extreme weather event.

Other area which is potentially threatened is a housing estate directly at basin 6. The block of flats lying parallel to the street

has an underground parking and during the flash flood in 2009 it was indeed inundated, and three fire-fighting squads were working through the night. What is worth mentioning here, this building was built in 2007.In 2010, a new housing estate was being built between basin 6 and 10, and by now is fully functional. What is more, both, block of flats and the hospital have been floded in a flash flood event in 2012 (Wyborcza, 2012). This example shows, that there is a need to explore flash floods as a reasonable threat to the urban areas, and the research should also aim at changing the urban planning policies. Fluvial floods

have existed in European planning policies for a long time and certain limitations regarding building localisations have been introduced. It is our strong belief, that pluvial floods in urban areas should earn the same publicity.

As a result of implemented model, the vulnerability map for the study area was derived. Model has to be verified, therefore following validation methods are recommended: a) IMGW supplied analysis (including radar data); b) hydrological modelling (i.e. MIKE URBAN or SOBEK) c) in situ measurements. The analysis supplied by IMGW as well as in situ measurements are

relatively cost-expensive (i.e. IMGW analysis of a short event on a small scale valued in thousands of euro). Therefore, for the purpose of this project, we propose to validate the results based on the outcome of hydrological modelling. These have been widely used, also for urban catchments, and are commonly used in many European countries.

Also, in order to improve the results, rainfall data has to be obtained n the better resolution, both spatial and temporal. The only available data was provided by IMGW from one meteorological station Warszawa - Okecie, which is approximately 8

km remote from the study area. It resulted in the assumption that there is the same rainfall input on each m$^3$ or rather raster cell. As mentioned above, the analysis dealienated 1864 local catchments and identified the lowest points within them. Based on the surface feature and accumulated water volume, it was possible to asses how high the water would rise from the lowest point. It enabled to derive potentially vulnerable areas to the flash flood with the precipitation input of 39 mm/m$^2$. During the analysis, some part of rainfall was removed due to the soil infiltration and sewage system (theoretically, as in Verlaat et al.

(2015)), which directly influenced the accumulation value.

# 6  Conclusions and discussion

Flash floods in urban areas concern local municipalities, which often do not share interest to have an insight into the threats, which these floods may pose. It can be seen in many examples when extreme rainfall occurs, sudden and expensive measures



are taken to mitigate its negative effects, instead of planning to prevent such occurrences. The key to the properly managing such threats lies in anticipation and there are of course many ways to achieve that. Many researches (among others Ochoa-Rodriguez et al. (2015) or Ntegeka et al. (2015)) recently focus on i.e. how to detect these rainfall events on such a small area which a urban catchment is or how to warn municipalities and rise emergency effectiveness in case of flash flood (Einfalt et al.,

2009). Many advances have been made recently in obtaining high resolution precipitation at small, urban scales. Nevertheless, some studies argue, that currently available rainfall measurements might still be too coarse to properly model urban hydrology (Ochoa-Rodriguez et al., 2015). Focusing on forecasting heavy rainfall and flash floods is equally important as modelling these events. For many years now hydrological sciences focus on these, also in urban areas, and some models are implemented nation wide (Banasiak, 2012) and treated as reference and calibration for other models. They provide valuable insight into

processes happening during the flash floods. Nevertheless, in order to deliver satisfactory results, they require a lot of detailed and various input data and computing time. For that reason, we propose an alternative to this modelling, with GIS based simplified methodology to detect urban areas especially vulnerable to the flash floods. This means to balance the accuracy, calculating time and input data needed. The case study shows, that proposed methodology finds jeopardised locations and thus this accuracy is adequate to identify problems in case of violent rainfall event. What is more, the computation time is faster and

data input not as robust as in case of complicated hydrological models. Such a model should be used to determine urban areas which might prove problematic to manage them better in the future i.e. to include them in local zoning or site plans. There have already been some similar studies carried out across the Europe, like the studies LISEM De Roo et al. (1996), WOLK (Klok, 2012) or SOBEK (van Dijk et al., 2014) or as well as city projects (Bremen, 2015), however our main goal is to create a simple, open-source platform to be available for usage in every municipality. In further studies, the difference and sensitivity

to the input data between developed methodology and above mentioned studies will be investigated.

For the further analysis we propose some improvements. The very first step is now creating the DTM with higher resolution (derived from LiDAR data). What is more, the developed model has not been calibrated due to lack of proper data from IMGW. Therefore, we propose, that in further studies hydrological models are run (as in Hunter et al. (2008) or Fewtrell et al. (2011)) in the same area with the same input data, and results will be compared. It is known, that models should not be calibrated

based on other models, as argued in Gupta et al. (1998), however in such case, required in situ measurements greatly exceed available funds. Furthermore, as mentioned above, the analysis was based on the presumption of equal rainfall with no temporal resolution (daily summary) and one meteorological station 8 km away. We will pursue in the future to obtain data from IMGW from radar measurements and try to implement approach suggested in Ochoa-Rodriguez et al. (2015) to merge radar-rain gauge measurements for urban applications. Last, but not least, in further work we will include sewage system and storm

inlets into the analysis. As we aim to develop a consistent methodology, which could be implemented in many places, we are struggling now to extract the vital information from these datasets automatically. City district Bergen-Enkheim in Frankfurt, the second test area, was chosen due to severe damages it underwent as a result of flash floods (Rundschau, 2016). Until now aerial orthophotomaps (RGBI) with the resolution of 20 cm, taken at 18.07.2014, DTM, geodatabase of the land cover for the area from 01.06.2015 and sewage system in CAD were obtained. Here as well we were confronted with the challenge which

transforming the drawing into an actual, georeferenced geodatabase entry means. The radar and rain-gauged data are expected



to be delivered soon from DWD (Deutscher WetterDienst) for the extended period (2008-2016) in order to derive extreme weather events, but also to study the overall trend. We are currently working on the implementation of the methodology in the second test area and the differences between available data in these two municipalities (including sensitivity of the model to the different input data) are thoroughly analysed.

5     Understanding the mechanisms acting during flash floods in urban areas is important not only for scientific purposes, but also for local communities and politicians. It is reasonable to believe, that as global temperatures rise, local climates will become more unpredictable (Halder et al., 2015) due to deforestation, urbanisation, rapid population growth and many other man-made pressures. Therefore, extreme rainfall events and flash floods will become more prevalent in Europe climate zone and will cause more and more damage, especially in urban areas. Although proposed methodology skips many factors, as opposed to

10 hydrological modelling, it allows to quickly identify vulnerable areas to the effects of flash floods, which may lead to the better management of these areas prior to the weather event. This particular approach enables to delineate endangered areas based on the data that great majority of European municipalities should have available, thus could be able to perform the analysis on their own. Although the accuracy of the presented methodology is much coarser than one delivered by the hydrological models, it is based on open-source software and does not require the expert knowledge to run it. Since the increase in weather

15 events is already noticeable, more studies should be focused on researching this topic.



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
