# Peer review of "Identifying urban areas prone to flash floods using GIS - preliminary results"

_Hydrology and Earth System Sciences, 2016_

## Referee Comment (RC1) · Anonymous Referee #1 · 28 Nov 2016

General comments:

This paper aims to create an easy-to-use methodology for identifying the urban areas that are particularly vulnerable to pluvial floods. Such an attempt is of significance to both scientific understanding of and practical management of pluvial flooding. However, the current version of this paper is not convincing in terms of scientific significance, scientific quality, and presentation quality, although the authors indeed developed a sound proposal and may achieve the accomplishment in the near future. A reject decision is therefore suggested and simultaneously authors are encouraged to continue improving this work.

Specific comments:

[Figure]

1. The methodology and results were not calibrated and validated. As authors mentioned, calibration and validation are expensive and difficult; however, they are still indispensable for a convincing research. If it is difficult to obtain the in-situ measurements, hydrological/hydraulic model results, and aerial photograph, household survey based on questionnaire may be an alternative tool to calibrate and validate the study.

2. I do not think the spatial variation in precipitation is a big problem for a study area of 327 ha. In contrast, a short-interval precipitation (one hour or less) should be applied for identifying urban flooding areas.

3. As the clarified aims of this paper is a new methodology, I suggest possible revisions should be focused on the methodology part. It was difficult to find the strong points of the methodology based on the current version.

4. I suggest a clarified and well-structured presentation in future revision. For example, the Result section contained too many points that should be in Discussion section. It is nice to see 10 figures in a paper; however most of these figure only depicted very general information.

---

## Referee Comment (RC2) · Anonymous Referee #2 · 30 Nov 2016

General comments:

The purpose of the article is to develop a method to identify urban areas vulnerable to flash flooding. This falls within the scope of HESS. The presented line of research has potential as it pursues a versatile methodology, applicable to multiple European cities, without high modelling or expertise demands.

However, the submitted article is not at a publishable level. It lacks structure and legibility. Presented methods are not novel and results were not validated. Tackling this issues requires more than a major revision. This prompts me to recommend the article rejection.

I take the opportunity to encourage the authors to continue with this line of thought as it can certainly render substantial results in the future. An article describing the mining of INSPIRE-compliant data sets, and other open data describing flooding impacts, to automatically identify and validate flood prone areas in a batch of European cities, would yield a novel scientific contribution.

Specific comments:

1. The article does not present a substantial contribution to current knowledge or techniques in urban flooding. Similar methods have been previously used and are available in related literature. The work of van Dijk et al. (2013), cited by the authors, is a recent example of an already published work in this topic. That paper uses the same method with a closely similar purpose in an urban environment. However, it does include a basic visual comparison of results achieved by the D8 modeling and by a 1D-2D hydraulic simulation. Such comparison is lacking in the article under revision.

2. Even though the rationale behind the applied methods is valid, obtained results are not discussed in sufficient detail. The validity of results cannot be determined with the information delivered in the article. The authors do mention that the areas modeled as flood-prone included a hospital and a domestic complex that were previously affected by floods. However this textual description is not sufficiently detailed to afford a proper evaluation of the method performance. A comparison between modeled areas and impact data, delivering quantitative performance metrics, could leverage the scientific potential of this work.

3. The structure of the article requires a major revision. For instance, sentences consisting of discussions are often found in Introduction and Results sections. The information in tables and figures is not used or discussed in detail, or its purpose is unclear. The use of English language (style, word choice, punctuation) should be also carefully revised.

---

## Referee Comment (RC3) · Anonymous Referee #3 · 5 Dec 2016

The paper aims at presenting a simplified methodology to identify urban areas prone to urban flooding due to intense precipitation.

In my opinion, this paper does not present novel approaches in the fields of GIS or pluvial flood modelling. As the authors present in the paper, the GIS methodologies used are well known and available at most of GIS software and the methodology presented lacks some of the advanced hydraulic analysis essential to fully understand the hydraulic behaviour of urban drainage systems.

Throughout the paper the authors identify some limitations of the presented study, e.g. the quality of the data sets is not adequate (e.g. too coarse) (lines 1-4 of Page 9) and the sewer network and the dynamics of urban drainage systems (especially

during flooding events) is not considered (Line 16 in Page 16). The authors seem aware of these limitations as they point various possibilities for further investigations. They should include these further investigations in the paper and, for example, use them as benchmark to assess the accuracy of the results obtained using the presented "simplified" methodology.

The methodology requires the use of a value of precipitation (in this case 39 mm/m2 - Page, line 28). Maybe the authors can relate this to a precipitation return period? of course the methodology does not consider the precipitation duration, but perhaps the concept of return period could be considered in the methodology!? 39 mm/m2 shall cause no relevant flooding issues if it happens during the time period of one month (see Tab 2 in page 8).

The structure of the paper must be revised, e.g. Section 1.2. seems too long; is all this information relevant to fully understand the paper?

Figure 4 caption is not related to the presented figure and is the same as Figure 6. Also, the quality (e.g. resolution and size) of the figures may be improved.

The level of English shall be improved: minor typos and major grammar deficiencies make the paper difficult to read.
* * *

---

## Short Comment (SC1) · 12 Dec 2016

This manuscript describes the modelling of areas prone to flash flooding in a district of Warsaw, Poland using GIS. The general concept of the research was interesting and could potentially be widely applicable and useful in various cities. A relatively strong case was built for modelling flood-prone areas, particularly with the advent of climate change. The idea of including open source data and/or open source GIS platforms is helpful as it allows a broader audience of GIS users to map the potential for flood risk or other environmental risks. These points notwithstanding, there were two major, interconnected issues with the article – weak connection between purpose, objectives and methods as well as the utility of results.

a) Weak Connection between Purpose, Objectives and Methods The article does not clearly connect the purpose, objectives and methods because the purpose is unclear, the objectives appear unfulfilled and the methods do not align with the purpose or objectives. It must be clear to the reader whether the primary focus of the article is to capitalize on the use of open source GIS data and/or software or if this is merely a possibility. If the focus of the article is on open source GIS data or open source GIS platforms, or both, then it should be mentioned explicitly in the title and abstract. More information on the history and benefits of open source would be helpful to allow the reader to understand why open source is important for modelling potential risk. The difference between open source data and open source platforms should also be made explicit for readers unfamiliar with the realm of open source and GIS. The purpose appears to be an assessment of areas vulnerable to floods using open source GIS data and/or software, but the title, abstract and introduction do not make this clear. The purpose is not fully articulated and there is confusion throughout the article about what the purpose of the research is. Modelling flood-prone areas with open source data and proprietary GIS and modelling flood-prone areas with open source data and open source GIS are two very different purposes. The abstract mentions "commonly available data", but the introduction refers to "open data", which do not necessarily share the same definitions. Additionally, it is unclear whether the model used in this research is an open source or proprietary platform. The introduction referred to the use of an open-access GIS platform, but then the conclusion referred to the creation of an open-source platform to model flood risk. It is not relevant to mention the use of open-access GIS platforms in the introduction if the project does not use this type of GIS platform. The objectives of the research seem to confuse the research undertaken with the research that could be undertaken in the future. Steps b) through d) of the research (lines 28-30 on p. 2) do not appear to be addressed; however, there is reference to these steps in the conclusion section as research that will be conducted. For instance, step b) of the research section is to "test the thesis in different European urban areas", but it states later in the paragraph that the paper tests the functionality

of flood-prone area modelling in Warsaw. The objectives of the research should be attainable; testing the modelling in different European urban areas is not a realistic objective of this research. The methods appear to need more assessment for feasibility and there seems to be an overemphasis on methods that were not applicable to this research. Parts of the methods that did not work should be excluded from the paper (e.g. lines 12-16 on p. 12 and lines 1-3 on p. 13), unless they were appropriately adjusted and yielded meaningful results. Further, a large part of the introduction is dedicated to deriving the digital surface model from LiDAR data, though LiDAR data was unavailable in the study area at the time. Again, the focus of the paper should be on research that was conducted and not on research that could be conducted. It would be more suitable to acknowledge the current availability of LiDAR data in the conclusion as an aspect that could improve the accuracy of the model. There should be a coherent narrative in which the purpose feeds into the objectives (how purpose will be fulfilled), which feed into the methods (how objectives will be fulfilled). This will ultimately be the reader's guide to understanding what the research is, its importance and how it will be conducted. b) Utility of Results The results could have been more effective if better methods were chosen. A major issue with the results was that it only modeled one possible scenario when flooding events are highly dynamic. The highest maximum daily precipitation from the summers of 2007-2016 was chosen to assess flood-prone risk, but it would have been more valuable to model various scenarios with 25-, 50-, 100- and 200-year precipitation events to identify the areas at risk. This would provide insight into how flood waters rise across space and time since flood waters do not necessarily rise uniformly. The analysis of these four scenarios could also be conducted quickly. Moreover, the results were never validated with data, so it is difficult to assess the accuracy of the model. It would be useful to attempt reconciling the model data with the 2009 flash flood event data. Validating the model is an important step to understanding and addressing the model's shortfalls. Alternatively, it may be suitable to create a "risk index" using multi-criteria analysis where different layers of data can be used to assess cumulative risk.

General Suggestions for Improvement c) Review correct use of citations (e.g. lines 4-5 on p. 2 do not need brackets around the years), quotations (e.g. lines 3-4 on p. 11) and figure references (e.g. do not use 'see' at line 32 on p. 11) d) Solicit the help of a few proofreaders as there were a few instances where the wrong word was used (e.g. p. 11 engendered vs. endangered) or where the word was spelled incorrectly (e.g. p. 9 topolgic vs. topologic). e) Remove table of LiDAR data availability in selected European countries since it does not pertain to research. f) Ensure that sentences are directly relevant to the research. It is not important to include the types of houses or industrial nature of the city unless an explicit link is made to flood risk (e.g. p. 5). g) Use only aerial photograph from 2008 in figure 1 as the research is to model flood-prone areas during the time of the research, and not flood-prone areas in 1935 or 1945. h) Number steps for methods to make it easier to follow. i) Explain why the total amount of water accumulated in each basin was classified into 5 classes with the top 3 classes being the most risky. There should be a rationale behind this. j) Enlarge figure 10 so the reader can easily identify basin classification and flooded areas. k) Separate discussions from conclusion. The discussion should relate directly to the objectives and limitations while the conclusion should give an overview of the research, its importance, findings and implications.

I would recommend reflecting on the purpose, objectives and methods and narrowing these elements. In its current state, the paper is difficult to follow and the results leave a little to be desired. Once it is clear what the purpose of the article is, the appropriate objectives and methods can be chosen to support it. A key question is whether the focus of the research is on open source data and/or open source platforms and whether the data is available to support this research. There should also be a very critical review of whether every sentence in the paper is relevant to the research. I believe major revisions are necessary to strengthen the utility of this research.

---

## Short Comment (SC2) · 16 Dec 2016

A Review of "Identifying urban areas prone to flash floods using GIS – preliminary results" – by Marzena Wicht and Katarzyna Osinka – Skotak

This paper proposes a modeling methodology for the identification of urban areas that are vulnerable to flash floods, and classifying them based on their level of risk. The paper has a very practical application in the world today, and may help decision makers better develop and protect their citizens and infrastructure. However, the paper requires much rewriting, clarification, and more detailed information regarding what it is that makes it innovative for it to be accepted. Please see my suggestions for how to improve to paper, as well as, suggestions on how to improve the wording.

[Figure]

1) While reading this paper, I found many adjectives that are unnecessary, and other adjectives that may be very descriptive but with no descriptions. For example:

a. P.1-L.2 "consistent methodology" scientific methods are consistent by nature.

b. P.1-L.2 "particularly vulnerable" I understand that the particularities may be explained later, but I believe a more descriptive wording would be beneficial.

c. P.1-L.7 "torrential rains" The word torrential does not have a specific measurement itself, so if it is used you may wish to elaborate on the intensity of this rainfall.

d. P.1-L19 "violent weather" Like torrential, it is important for the reader to understand the meaning of violent. Although I understand you are citing from Gaume et al. 2009, however they do not use the word violent.

e. P1.1-L24 "valid threat" validation of a threat may have to be discussed if the word valid is used. I understand this word adds colour and emphasis to the statement, but simply saying it presents a threat to the inhabitants is good enough.

Spelling mistakes, and improper or awkward use of words are found throughout this paper and highly impact the way this very interesting paper is delivered.

2) The Introductory section introduces elements in an awkward fashion. I believe the whole section (and whole paper) should be restructured to clearly introduce the flood risk mitigation modeling, its limitations, and opportunities. I believe introducing sub-chapter sections and clearly explain the relevance of each should be easily done, as this paper indicates a high level of thought and research has already been put into it. Here is my proposal using some of the themes, and information you spoke of in the introduction. This is not necessarily the best way to present this paper, but I believe something along these lines will greatly improve the foundation for which you will be producing a methodology.

1. Introduction 1.1 Changing Weather 1.2 The nature of Urban centers 1.3 Availability of Data and Data Processing 1.4 Elements and Indicators That Influence Vulnerability

and Risk 1.5 Modeling of Flash Floods in Urban Areas

3) P.2-L13 which open-access (GIS) platform are you speaking of? Is it called GIS? A graphical information system (GIS), is a system, and there are many platforms that utilize standardized data structures, QGIS for example. Perhaps it is that municipalities have open-access GIS data available. This needs to be clarified along with P.2-L.15 "GIS tools".

4) I think you should introduce all elements of chapter 2 "Data and methods" into the introduction. Most of the information here is introductory into the advancement and availability of data and technology. Some information in these parts can be placed later in chapter 3 "design of experiment and methodology". For example: P.9-L3 "during preliminary trials . . ." This information would go very well in the design/ development of the experiment.

5) Chapter 3 is well broken down into subsections, and contains some very good information. However, it often lacks the justification and emphasis to achieve the goal of producing a generalized methodology. I believe you can easily fix this by clearly explaining each possible input and their short comings. It may be helpful for the sub-chapter headings to be the same as those in the model design figure headings, and explain each step by step this way.

6) You mention anthropogenic soil, and its importance to hydrology, I believe further emphasis can be placed on this, to explain how it is incorporated into the methodology as well as its sensitivity analysis on the results.

7) The first sentence of the conclusion should be like the first statement of the introduction. Something like "A methodology for identifying at risk areas has been accomplished.".

The Sentence (P.16-L.32) "Flash floods in urban areas. . ." is poorly written and introduces information not spoken about during the paper (social aspects regarding hydrological studies). If this needs to be included, it should be introduced properly within the introduction section, and refer to certain Socio-Hydrology publications (Like Wheater, Montanari, or others) I believe it could be elaborated on when discussing the availability of data and data processing.

8) Identifying what makes the approach presented innovative. The methodology design seems appropriate as argued. I believe it can be elaborated upon, further utilization of the works of model creation and analysis from Gupta's work.

9) The addition of storm water infrastructure into the model will add a level of innovation and is worth mentioning in the design methodology. This may be the most innovative and worth elaborating on.

---

## Author Comment (AC1) · 15 Jan 2017

Marzena Wicht and Katarzyna Osinska-Skotak

mwicht@gik.pw.edu.pl

RC = Reviewer comment AR = Authors reply

RC: General comments: This paper aims to create an easy-to-use methodology for identifying the urban areas that are particularly vulnerable to pluvial floods. Such an attempt is of significance to both scientific understanding of and practical management of pluvial flooding. How- ever, the current version of this paper is not convincing in terms of scientific signifi- cance, scientific quality, and presentation quality, although the authors indeed devel- oped a sound proposal and may achieve the accomplishment in the near future. A reject decision is therefore suggested and simultaneously authors are encouraged to continue improving this work.

[Figure]

AR: On behalf of all of the authors, I'd like to thank the rwviewer for the time spent on analysis of the manuscript

RC: 1. The methodology and results were not calibrated and validated. As authors men- tioned, calibration and validation are expensive and difficult; however, they are still indispensable for a convincing research. If it is difficult to obtain the in-situ measurements, hydrological/hydraulic model results, and aerial photograph, household survey based on questionnaire may be an alternative tool to calibrate and validate the study.

AC: As mentioned in the manuscript, for the time being, the validation could not be conducted due to insufficient data. We stressed however, that those are merely preliminary results and the validation process based on 1D/2D hydrological model is planned for further research.

RC: I do not think the spatial variation in precipitation is a big problem for a study area of 327 ha. In contrast, a short-interval precipitation (one hour or less) should be applied for identifying urban flooding areas.

AC: Thank you for the suggestion. The data, which we were able to acquire prior to the study were simply daily summaries of precipitation from rain gauges (hourly steps, also rain gauges, measured automatically, not verified and not stored in the central database of Polish Metrological Institute are not available free of charge). We recently got a confirmation that radar data (1 hour step) might be available for scientific purposes and we are currently inquiring to obtain it.

RC: 4. I suggest a clarified and well-structured presentation in future revision. For example, the Result section contained too many points that should be in Discussion section. It is nice to see 10 figures in a paper; however most of these figure only depicted very general information.

AC: We appreciate reviewer's suggestions – in the revised version of the manuscript we will focus on presenting strong points of the methodology and we will divide the

manuscript better. We also will decrease number of pictures to those essential ones.

---

## Author Comment (AC2) · 15 Jan 2017

Marzena Wicht

RC = Reviewer comment AR = Authors reply

RC:

The purpose of the article is to develop a method to identify urban areas vulnerable to flash flooding. This falls within the scope of HESS. The presented line of research has potential as it pursues a versatile methodology, applicable to multiple European cities, without high modelling or expertise demands. However, the submitted article is not at a publishable level. It lacks structure and leg- ibility. Presented methods are not

novel and results were not validated. Tackling this issues requires more than a major revision. This prompts me to recommend the article rejection. I take the opportunity to encourage the authors to continue with this line of thought as it can certainly render substantial results in the future. An article describing the mining of INSPIRE-compliant data sets, and other open data describing flooding impacts, to automatically identify and validate flood prone areas in a batch of European cities, would yield a novel scientific contribution.

AR:

We appreciate reviewer's valuable and constructive advice on our manuscript. We intend to implement suggested changes to improve the quality of this article. We thank as well for the relevant suggestion of INSPIRE-compliant data-sets, which definitely will enrich this research and will be included in the revised version of the manuscript.

RC:

Specific comments: 1. The article does not present a substantial contribution to current knowledge or techniques in urban flooding. Similar methods have been previously used and are available in related literature. The work of van Dijk et al. (2013), cited by the authors, is a recent example of an already published work in this topic. That paper uses the same method with a closely similar purpose in an urban environment. However, it does include a basic visual comparison of results achieved by the D8 modeling and by a 1D-2D hydraulic simulation. Such comparison is lacking in the article under revision.

AR:

Ad.1 We should like to thank the reviewer for these valuable comments. Although work of Van Dijk (2013) might bear similarities, it merely describes differences between simple surface flow model and 1D/2D hydrological modelling. What is proposed in this manuscript is enhanced surface flow model (coupled in the future research with the minor system) considering much more than just DEM.
RC:

2. Even though the rationale behind the applied methods is valid, obtained results are not discussed in sufficient detail. The validity of results cannot be determined with the information delivered in the article. The authors do mention that the areas modeled as flood-prone included a hospital and a domestic complex that were previously affected by floods. However this textual description is not sufficiently detailed to afford a proper evaluation of the method performance. A comparison between modeled areas and impact data, delivering quantitative performance metrics, could leverage the scientific potential of this work.

AR:

Ad.2 In the revised version of the manuscript we plan to discuss the results in the greater detail. In our future research we also plan to validate the results based on 1D/2D hydrological models, as well as perform sensitivity analysis (of the input data) to investigate their impact. RC: 3. The structure of the article requires a major revision. For instance, sentences consisting of discussions are often found in Introduction and Results sections. The information in tables and figures is not used or discussed in detail, or its purpose is unclear. The use of English language (style, word choice, punctuation) should be also carefully revised

AR: We thank for the suggestions – the manuscript will be revised in both – structure and writing style.

---

## Author Comment (AC3) · 15 Jan 2017

Marzena Wicht and Katarzyna Osinska-Skotak

mwicht@gik.pw.edu.pl

Marzena Wicht

RC = Reviewer comment AR = Authors reply

RC:

The paper aims at presenting a simplified methodology to identify urban areas prone to urban flooding due to intense precipitation. In my opinion, this paper does not present novel approaches in the fields of GIS or pluvial flood modelling. As the authors present in the paper, the GIS methodologies used are well known and available at most of GIS software and the methodology pre- sented lacks some of the advanced hydraulic anal-

ysis essential to fully understand the hydraulic behaviour of urban drainage systems. Throughout the paper the authors identify some limitations of the presented study, e.g. the quality of the data sets is not adequate (e.g. too coarse) (lines 1-4 of Page 9) and the sewer network and the dynamics of urban drainage systems (especially during flooding events) is not considered (Line 16 in Page 16). The authors seem aware of these limitations as they point various possibilities for further investigations. They should include these further investigations in the paper and, for example, use them as benchmark to assess the accuracy of the results obtained using the presented "simplified" methodology. The methodology requires the use of a value of precipitation (in this case 39 mm/m2 - Page, line 28). Maybe the authors can relate this to a precipitation return period? of course the methodology does not consider the precipitation duration, but perhaps the concept of return period could be considered in the methodology!? 39 mm/m2 shall cause no relevant flooding issues if it happens during the time period of one month (see Tab 2 in page 8). The structure of the paper must be revised, e.g. Section 1.2. seems too long; is all this information relevant to fully understand the paper? Figure 4 caption is not related to the presented figure and is the same as Figure 6. Also, the quality (e.g. resolution and size) of the figures may be improved. The level of English shall be improved: minor typos and major grammar deficiencies make the paper difficult to read.

AR:

On the behalf of the authors, I would like to thank the reviewer for all the comments and suggestions on our manuscript. In the revised version of it we will try to address all mentioned issues and include some (available) possibilities to overcome present limitations, which we can address now and couldn't back at the time when manuscript was submitted. However, the focus of this study was to utilize already available GIS tools in the hydrological field (contrary to hydrological models) with much simplified approach, where the lack of the advanced hydraulics analysis is not an obstacle to perform the analysis. In the revised version of the manuscript we will naturally address

the structure and writing style.

---

## Author Comment (AC4) · 15 Jan 2017

Marzena Wicht

RC = Reviewer comment AR = Authors reply

RC:

This manuscript describes the modelling of areas prone to flash flooding in a district of Warsaw, Poland using GIS. The general concept of the research was interesting and could potentially be widely applicable and useful in various cities. A relatively strong case was built for modelling flood-prone areas, particularly with the advent of climate change. The idea of including open source data and/or open source GIS platforms

is helpful as it allows a broader audience of GIS users to map the potential for flood risk or other environmental risks. These points notwithstanding, there were two major, interconnected issues with the article – weak connection between purpose, objectives and methods as well as the utility of results.

AR:

On the behalf of all authors, I would truly like to thank Ms Wong for all the work spent on the manuscript.We greatly appreciate the comments and acknowledge that the suggestions will lead to the improvement of the manuscript.

RC:

a) Weak Connection between Purpose, Objectives and Methods The article does not clearly connect the purpose, objectives and methods because the purpose is unclear, the objectives appear unfulfilled and the methods do not align with the purpose or objectives. It must be clear to the reader whether the primary focus of the article is to capitalize on the use of open source GIS data and/or software or if this is merely a possibility. If the focus of the article is on open source GIS data or open source GIS platforms, or both, then it should be mentioned explicitly in the title and abstract. More information on the history and benefits of open source would be helpful to allow the reader to understand why open source is important for modelling potential risk. The difference between open source data and open source platforms should also be made explicit for readers unfamiliar with the realm of open source and GIS. The purpose appears to be an assessment of areas vulnerable to floods using open source GIS data and/or software, but the title, abstract and introduction do not make this clear. The purpose is not fully articulated and there is confusion throughout the article about what the purpose of the research is. Modelling flood-prone areas with open source data and proprietary GIS and modelling flood-prone areas with open source data and open source GIS are two very different purposes. The abstract mentions "commonly available data", but the introduction refers to "open data", which do not necessarily

share the same definitions. Additionally, it is unclear whether the model used in this research is an open source or proprietary platform. The introduction referred to the use of an open-access GIS platform, but then the conclusion referred to the creation of an open-source platform to model flood risk. It is not relevant to mention the use of open-access GIS platforms in the introduction if the project does not use this type of GIS platform. The objectives of the research seem to confuse the research undertaken with the research that could be undertaken in the future. Steps b) through d) of the research (lines 28-30 on p. 2) do not appear to be addressed; however, there is reference to these steps in the conclusion section as research that will be conducted. For instance, step b) of the research section is to "test the thesis in different European urban areas", but it states later in the paragraph that the paper tests the functionality of flood-prone area modelling in Warsaw. The objectives of the research should be attainable; testing the modelling in different European urban areas is not a realistic objective of this research. The methods appear to need more assessment for feasibility and there seems to be an overemphasis on methods that were not applicable to this research. Parts of the methods that did not work should be excluded from the paper (e.g. lines 12-16 on p. 12 and lines 1-3 on p. 13), unless they were appropriately adjusted and yielded meaningful results. Further, a large part of the introduction is dedicated to deriving the digital surface model from LiDAR data, though LiDAR data was unavailable in the study area at the time. Again, the focus of the paper should be on research that was conducted and not on research that could be conducted. It would be more suitable to acknowledge the current availability of LiDAR data in the conclusion as an aspect that could improve the accuracy of the model. There should be a coherent narrative in which the purpose feeds into the objectives (how purpose will be fulfilled), which feed into the methods (how objectives will be fulfilled). This will ultimately be the reader's guide to understanding what the research is, its importance and how it will be conducted.

AR:

Ad. a) We acknowledge that the connection mentioned is indeed weak and we intend to address this matter in the revised version of the manuscript. We also appreciate the suggestion on reviewing the 'open source' research to include in the paper. Although we mentioned carrying out the research on the examples of different European cities – the preliminary results discussed in this manuscripts are the focus here. We will investigate this in our future research in other case studies. Since the submission of the manuscript we were also able to acquire some additional data (i.e. LiDAR based DTM), which we can now include in the revised version of the manuscript.

RC:

b) Utility of Results The results could have been more effective if better methods were chosen. A major issue with the results was that it only modeled one possible scenario when flooding events are highly dynamic. The highest maximum daily precipitation from the summers of 2007-2016 was chosen to assess flood-prone risk, but it would have been more valuable to model various scenarios with 25-, 50-, 100- and 200-year precipitation events to identify the areas at risk. This would provide insight into how flood waters rise across space and time since flood waters do not necessarily rise uniformly. The analysis of these four scenarios could also be con- ducted quickly. Moreover, the results were never validated with data, so it is difficult to assess the accuracy of the model. It would be useful to attempt reconciling the model data with the 2009 flash flood event data. Validating the model is an important step to understanding and addressing the model's shortfalls. Alternatively, it may be suitable to create a "risk index" using multi-criteria analysis where different layers of data can be used to assess cumulative risk.

AR:

We appreciate the suggestion regarding the various scenarios with 25-, 50-, 100- and 200-year precipitation events – we would like to include such scenarios in the revised version of the manuscript. The validation however is much harder, as analyses performed by Polish Metrological Institute (regarded as a benchmark i.e. during court trials) are not available free of charge. We want however, in our future research, to perform validation based on 1d/2d hydrological models. We are also considering the 'risk index' mapping, so thank you for confirming our assumptions.

RC:

c) Review correct use of citations (e.g. lines 4-5 on p. 2 do not need brackets around the years), quotations (e.g. lines 3-4 on p. 11) and figure references (e.g. do not use 'see' at line 32 on p. 11) d) Solicit the help of a few proofreaders as there were a few instances where the wrong word was used (e.g. p. 11 engendered vs. endangered) or where the word was spelled incorrectly (e.g. p. 9 topolgic vs. topologic). e) Remove table of LiDAR data availability in selected European countries since it does not pertain to research.

AR:

We will review the citations, figures and proofread the manuscript before resubmitting.

RC:

f) Ensure that sentences are directly relevant to the research. It is not important to include the types of houses or industrial nature of the city unless an explicit link is made to flood risk (e.g. p. 5). g) Use only aerial photograph from 2008 in figure 1 as the research is to model flood-prone areas during the time of the research, and not flood-prone areas in 1935 or 1945. h) Number steps for methods to make it easier to follow.

AR:

We will exclude the irrelevant information.

RC:

i) Explain why the total amount of water accumulated in each basin was classified into 5

classes with the top 3 classes being the most risky. There should be a rationale behind this. j) Enlarge figure 10 so the reader can easily identify basin classification and flooded areas. k) Separate discussions from conclusion. The discussion should relate directly to the objectives and limitations while the conclusion should give an overview of the research, its importance, findings and implications. I would recommend reflecting on the purpose, objectives and methods and narrowing these elements. In its current state, the paper is difficult to follow and the results leave a little to be desired. Once it is clear what the purpose of the article is, the appropriate objectives and methods can be chosen to support it. A key question is whether the focus of the research is on open source data and/or open source platforms and whether the data is available to support this research. There should also be a very critical review of whether every sentence in the paper is relevant to the research. I believe major revisions are necessary to strengthen the utility of this research.

AR:

We will explain the classification. And in general we intend to improve the structure of the whole manuscript.

---

## Author Comment (AC5) · 15 Jan 2017

Marzena Wicht

RC = Reviewer comment AR = Authors reply

RC:

A Review of "Identifying urban areas prone to flash floods using GIS – preliminary results" – by Marzena Wicht and Katarzyna Osinka – Skotak This paper proposes a modeling methodology for the identification of urban areas that are vulnerable to flash floods, and classifying them based on their level of risk. The paper has a very practical application in the world today, and may help decision makers better develop and

protect their citizens and infrastructure. However, the paper requires much rewriting, clarification, and more detailed information regarding what it is that makes it innovative for it to be accepted. Please see my suggestions for how to improve to paper, as well as, suggestions on how to improve the wording.

AR:

We would like to thank for all the suggestions on the work and acknowledge that they will lead to the improved version of the manuscript.

RC: 1) While reading this paper, I found many adjectives that are unnecessary, and other adjectives that may be very descriptive but with no descriptions. For example: a. P.1-L.2 "consistent methodology" scientific methods are consistent by nature. b. P.1-L.2 "particularly vulnerable" I understand that the particularities may be explained later, but I believe a more descriptive wording would be beneficial. c. P.1-L.7 "torrential rains" The word torrential does not have a specific measurement itself, so if it is used you may wish to elaborate on the intensity of this rainfall. d. P.1-L19 "violent weather" Like torrential, it is important for the reader to understand the meaning of violent. Although I understand you are citing from Gaume et al. 2009, however they do not use the word violent. e. P1.1-L24 "valid threat" validation of a threat may have to be discussed if the word valid is used. I understand this word adds colour and emphasis to the statement, but simply saying it presents a threat to the inhabitants is good enough. Spelling mistakes, and improper or awkward use of words are found throughout this paper and highly impact the way this very interesting paper is delivered.

AR:

All the misused and inappropriate words throughout the manuscript, as well as spelling mistakes will be proofread and revised.

RC:

2) The Introductory section introduces elements in an awkward fashion. I believe the

whole section (and whole paper) should be restructured to clearly introduce the flood risk mitigation modeling, its limitations, and opportunities. I believe introducing sub-chapter sections and clearly explain the relevance of each should be easily done, as this paper indicates a high level of thought and research has already been put into it. Here is my proposal using some of the themes, and information you spoke of in the introduction. This is not necessarily the best way to present this paper, but I believe something along these lines will greatly improve the foundation for which you will be producing a methodology. 1. Introduction 1.1 Changing Weather 1.2 The nature of Urban centers 1.3 Availability of Data and Data Processing 1.4 Elements and Indicators That Influence Vulnerability and Risk 1.5 Modeling of Flash Floods in Urban Areas

AR:

We greatly appreciate the structure suggestions – we will definitely utilize them in the revised version of the manuscript

RC:

3) P.2-L13 which open-access (GIS) platform are you speaking of? Is it called GIS? A graphical information system (GIS), is a system, and there are many platforms that utilize standardized data structures, QGIS for example. Perhaps it is that municipalities have open-access GIS data available. This needs to be clarified along with P.2-L.15 "GIS tools".

AR:

As it was pointed out by other reviewers, the emphasis put on the open source GIS is missing – we plan to address this matter and explain it.

RC:

4) I think you should introduce all elements of chapter 2 "Data and methods" into the introduction. Most of the information here is introductory into the advancement and availability of data and technology. Some information in these parts can be placed

later in chapter 3 "design of experiment and methodology". For example: P.9-L3 "during preliminary trials" This information would go very well in the design/ development of the experiment. 5) Chapter 3 is well broken down into subsections, and contains some very good in- formation. However, it often lacks the justification and emphasis to achieve the goal of producing a generalized methodology. I believe you can easily fix this by clearly explaining each possible input and their short comings. It may be helpful for the sub- chapter headings to be the same as those in the model design figure headings, and explain each step by step this way. 6) You mention anthropogenic soil, and its importance to hydrology, I believe further emphasis can be placed on this, to explain how it is incorporated into the methodology as well as its sensitivity analysis on the results

AR:

We would like to use your suggestions in the revised version of the manuscript.

RC:

7) The first sentence of the conclusion should be like the first statement of the intro- duction. Something like "A methodology for identifying at risk areas has been accom- plished.". The Sentence (P.16-L.32) "Flash floods in urban areas..." is poorly written and intro- duces information not spoken about during the paper (social aspects regarding hydrological studies). If this needs to be included, it should be introduced properly within the introduction section, and refer to certain Socio-Hydrology publications (Like Wheater, Montanari, or others) I believe it could be elaborated on when discussing the availability of data and data processing. 8) Identifying what makes the approach presented innovative. The methodology design seems appropriate as argued. I believe it can be elaborated upon, further utilization of the works of model creation and analysis from Gupta's work. 9) The addition of storm water infrastructure into the model will add a level of innovation and is worth mentioning in the design methodology. This may be the most innovative and worth elaborating on.

AR:

Thank you for all your constructive comments – we will revise the social and utilization aspects (endorsed by the literature research) and will try to address all other issues which emerged during the review.

---

## Author Comment (AC6) · 24 Jan 2017

Dear Reviewers, Dear Editor,

I would like to take this opportunity, and thank once again for all the comments and suggestions you made on our manuscript. We appreciate them greatly and we will work on implementing them in the revised version of the manuscript. Since we submitted the manuscript for the discussion, we have worked constantly to improve its quality by overcoming the limitations we mentioned in it.

Consequently, we were able to acquire more input data, which leads to better results. We obtained i.e. LiDAR data for our case study area, as well as almost 10 years

precipitation data from 6 metrological stations in Warsaw. These two directly affect the results and we plan to utilize and discuss them in our revised manuscript. Taking all your comments into consideration, we plan to:

- restructure and clarify our presentation,

- focus on the strong points of the methodology by reinforcing purpose, objectives and methods used,

- include discussion of the INSPIRE – compliant data sets as possible input data,

- strengthen the 'open source' part of the study,

- discuss in much greater detail the results, their spatial variability and explain the classification,

- revise the language (style, word choice and punctuation) with academic American native speaker.

However, I would like to stress once again, that the focus of this study was to utilize already available GIS tools in the hydrological field (contrary to hydrological models) with much simplified approach, where the lack of the advanced hydraulics analysis is not an obstacle to perform the analysis. Furthermore, the title states 'preliminary results' which obviously suggests our tireless work on achieving better results in our future studies. I know that your reviews have contributed greatly to improve the quality of this work, and if given the chance, we would like to present their outcome in the revised version of the manuscript.

On behalf of my co-author and myself,

Marzena Wicht